# Silencing of TGFβ signalling in microglia results in impaired homeostasis

Tanja Zöller[1,2], Artur Schneider[1], Christian Kleimeyer[1], Takahiro Masuda[3], Phani Sankar Potru[2,4], Dietmar Pfeifer[5], Thomas Blank[3], Marco Prinz[3,6] & Björn Spittau[1,4]

TGFβ1 has been implicated in regulating functional aspects of several distinct immune cell populations including central nervous system (CNS) resident microglia. Activation and priming of microglia have been demonstrated to contribute to the progression of neurodegenerative diseases and, thus, underlie stringent control by endogenous regulatory factors including TGFβ1. Here, we demonstrate that deletion of *Tgfbr2* in adult postnatal microglia does neither result in impairment of the microglia-specific gene expression signatures, nor is microglial survival and maintenance affected. *Tgfbr2*-deficient microglia were characterised by distinct morphological changes and transcriptome analysis using RNAseq revealed that loss of TGFβ signalling results in upregulation of microglia activation and priming markers. Moreover, protein arrays demonstrated increased secretion of CXCL10 and CCL2 accompanied by activation of immune cell signalling as evidenced by increased phosphorylation of TAK1. Together, these data underline the importance of microglial TGFβ signalling to regulate microglia adaptive changes.

[1] Institute for Anatomy and Cell Biology, Department of Molecular Embryology, Faculty of Medicine, University of Freiburg, 79104 Freiburg, Germany. [2] Faculty of Biology, University of Freiburg, 79104 Freiburg, Germany. [3] Institute of Neuropathology, Medical Center—University of Freiburg, Faculty of Medicine, University of Freiburg, 79106 Freiburg, Germany. [4] Institute of Anatomy, University of Rostock, 18057 Rostock, Germany. [5] Department of Hematology/Oncology, Faculty of Medicine, University of Freiburg, 79106 Freiburg, Germany. [6] BIOSS Centre for Biological Signalling Studies, University of Freiburg, 79104 Freiburg, Germany. Correspondence and requests for materials should be addressed to B.S. (email: bjoern.spittau@med.uni-rostock.de)

Microglia represent the specific immune cell population of the central nervous system (CNS), are involved in physiological CNS functions and further participate in the development, progression as well as the resolution of pathological conditions[1,2]. As part of the innate immune system, microglia are able to sense pathogen-associated molecular patterns and danger-associated molecular patterns resulting in acute microglia reactivity[3]. However, chronic stimulation of microglia has been shown to induce a hyper-reactive phenotype which is referred to as primed and which is characterised by prolonged and exacerbated reactions. Primed microglia have been detected in several neurodegenerative diseases and during aging of microglia[4,5]. Recent studies have demonstrated that microglia derive from primitive macrophage precursors[6] involving PU.1- as well as IRF8-dependent signalling pathways[7]. Homing of microglia towards the CNS parenchyma is mediated by the CSFR1/IL-34 receptor/ligand pair and occurs during mid and late embryonic development[6,8]. Microglia develop a specific gene expression signature in the CNS parenchyma, which is distinct from the molecular signature of monocytes and monocyte-derived macrophages providing the possibility to segregate these distinct cell populations from each other[9,10]. The cell-surface protein TMEM119 was recently characterised as a microglia-specific marker in mouse and human and has been further used to determine that mouse microglia adopt their specific gene expression signature within the first two postnatal weeks[11]. This distinct microglial molecular signature was absent in mice with a CNS-specific deletion of transforming growth factor beta 1 (TGFβ1) and mutant mice displayed a reduction in microglia numbers which was observed from postnatal day 20 into adulthood. Moreover, postnatal development and survival of these mutant mice were impaired suggesting the contribution of mechanisms outside of the CNS to the phenotype development[12]. Nevertheless, it remains elusive whether the observed effects are dependent on microglial TGFβ signalling or whether secondary effects mediated by TGFβ-sensitive CNS cell populations, such as neurons, astrocytes or oligodendrocytes are responsible for the microglial phenotype reported by Butovsky et al.[12]. However, the above-mentioned results identified TGFβ1 as an essential endogenous factor to promote microglial maturation as a prerequisite for adequate microglia functions in the adult CNS. TGFβ1 has previously been described as a potent immunoregulatory factor for microglia in vivo and in vitro[13–15].

TGFβ-signalling is propagated by binding of TGFβ to the TGFβ receptor type II (Tgfbr2) that phosphorylates the TGFβ receptor type I (Tgfbr1), thus, triggering the kinase activity of Tgfbr1[16,17]. Receptor-associated signalling mediators Smad2 and Smad3 are the primary phosphorylation targets of Tgfbr1 and form a complex with the common mediator Smad4 to translocate to the nucleus and to further regulate the expression of TGFβ target genes[18].

Unfortunately, the postnatal lethality of TGFβ1-deficient mice due to severe systemic immune effects limited the analysis of the TGFβ1-mediated regulation of microglia activation in adult mice[19]. Moreover, lethal embryonic phenotypes of Tgfbr2- and Smad4-deficient mice restricted the evaluation of TGFβ-signalling in adult microglia[20,21]. Taking advantage of the unique microglial nature and the development of new genetic tools[22] we addressed the role of TGFβ-signalling during microglial maintenance and quiescence in adult mice.

Here, we used Cx3cr1Cre[ERT2] mice to induce the conditional deletion of the ligand-binding receptor Tgfbr2 in adult microglia. Our data indicate that postnatal silencing of microglial TGFβ-signalling impairs microglia quiescence rather than compromising microglia maintenance and survival in vivo. Moreover, RNAseq-based gene expression analysis revealed that silencing

TGFβ signalling in adult microglia does not affect the postnatally induced microglia-specific gene expression pattern but results in microglia activation and priming. We further demonstrate that increased TAK1 phosphorylation in Tgfbr2-deficient microglia and enhanced cytokine release could be detected suggesting the loss of microglial quiescence. Overall, these data underline the importance of TGFβ-signalling for regulation of microglia activation and further indicate that the maturation and maintenance functions of TGFβ1 take place at earlier postnatal stages. Finally, the mutant mouse line introduced in the present study constitute a powerful tool to analyse the contribution of microglial TGFβ signalling in neuropathological conditions associated with microglia activation.

## Results

**Induction of postnatal Tgfbr2 deletion in microglia.** In order to evaluate the importance of TGFβ-signalling for postnatal microglia survival and maintenance, a conditional microglia/macrophage-specific mutant mouse line for Tgfbr2 was generated by crossing Cx3cr1[CreERT2] mice to mice carrying loxP⁻site-flanked (floxed) alleles of Tgfbr2 to obtain Cx3cr1[CreERT2]:Tgfbr2[fl/fl] mice. The ROSA26-YFP (R26-yfp) mouse line was additionally crossed into the Cx3cr1[CreERT2]:Tgfbr2[fl/fl] line, resulting in the transgenic line Cx3cr1[CreERT2]:R26-yfp,Tgfbr2[fl/fl] (Fig. 1a). Tamoxifen (TAM)-induced recombination in vivo (Supplementary Fig. 1A) and in vitro (Supplementary Fig. 1B) resulted in the deletion of the floxed Exons 2/3 of the Tgfbr2 gene which code for the ligand-binding domain of the receptor (Fig. 1b). Sequencing of recombined primary microglia and qPCR after recombination in vitro and in vivo using exon-specific primers revealed deletion of Exons 2/3 in Cx3cr1[CreERT2]:R26-yfp,Tgfbr2[fl/fl] microglia (Supplementary Fig. 1). Noteworthy, lack of Exons 2/3 did not affect the reading frame of Tgfbr2 and, thus, recombined microglia expressed a dominant negative form of Tgfbr2 with normal serine/threonine kinase activity but abrogated ligand-binding capacity (Fig. 1b). In order to exclude phenotypic effects obtained due to Cx3cr1 haploinsufficiency in Cx3cr1[CreERT2] mice or TAM itself, additional control groups were used throughout this study. Cx3cr1[+/+]:R26-yfp,Tgfbr2[fl/fl] mice treated with TAM (+/+TAM) and Cx3cr1[CreERT2]:R26-yfp,Tgfbr2[fl/fl] mice treated with corn oil (cre/+OIL) were considered as controls, whereas Cx3cr1[CreERT2]:R26-yfp,Tgfbr2[fl/fl] mice treated with TAM (cre/+TAM) are referred to as recombined knockout mice in vivo. Similar control groups were used for in vitro experiments where ethanol served as the solvent for TAM (+/+EtOH, +/+TAM, Cre/+ EtOH, Cre/+ TAM). In order to test the recombination efficacy detection of stable YFP expression in microglia was assessed in vitro (Fig. 1c) and in vivo (Fig. 1d), as also described recently[22]. In both conditions, recombination rates of approximately 80% of F4/80⁺ microglia were achieved (Fig. 1c, in vitro 79.77 ± 6.69% YFP⁺ cells, Fig. 1d, in vivo 82.28 ± 18.53% YFP⁺ cells). Recombination efficacy was further analysed in CD115⁺/Ly6C[low] blood monocytes 1 week and 4 weeks after TAM injections to monitor the turnover of peripheral Cx3cr1⁺ myeloid cells. While a time-dependent reduction of YFP⁺ blood monocytes from 43% (1 week) to 9.8% (4 weeks) could be detected, the numbers of F4/80⁺/YFP⁺ microglia remained stable (Supplementary Fig. 2). Next, we validated the insensitivity of recombined microglia towards TGFβ1. Whereas recombinant TGFβ1 (5 ng/ml, 2 h) induced phosphorylation and nuclear translocation of SMAD2 in control microglia, no SMAD2 phosphorylation and nuclear translocation were detected in mutant microglia (Fig. 1e, f). Together, these data demonstrate that TAM-induced recombination resulted in the stable deletion of Tgfbr2 Exons 2/3 and abrogation of SMAD2-mediated TGFβ1 signal transduction in mutant microglia.

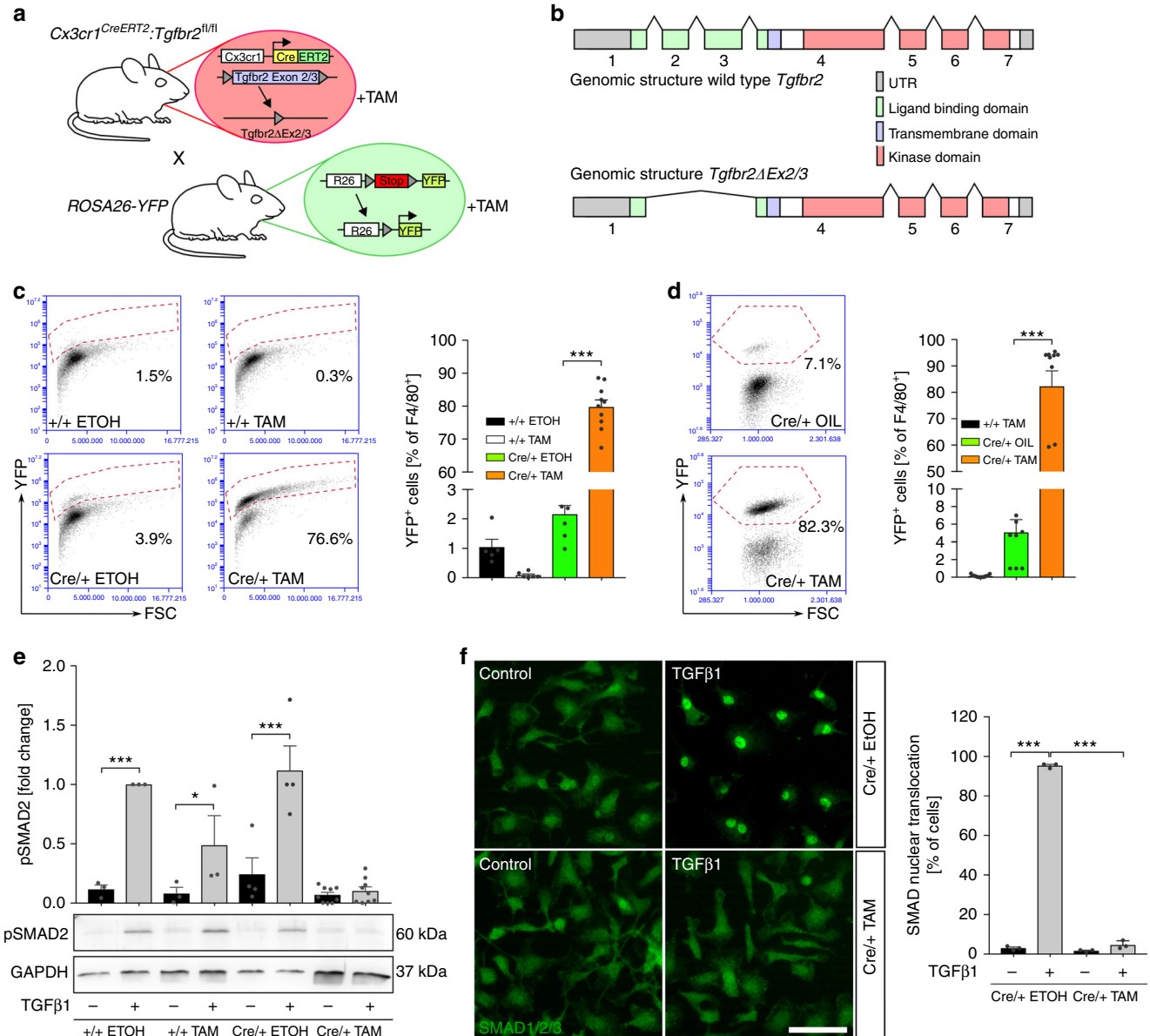

**Fig. 1** Successful microglia-specific knockout of Tgfbr2 in vitro and in vivo. **a** Scheme illustrating breeding strategy to obtain *Cx3cr1CreERT2:R26-yfp,Tgfbr2fl/fl* mice for TAM-induced deletion of *Tgfbr2* Exons 2/3 and induction of YFP reporter gene expression. **b** Genomic structure of wild-type and mutant *Tgfbr2* after TAM-induced recombination. **c** Flow cytometric analysis of microglia 3 days after TAM-induced recombination in vitro for expression of YFP. Data are presented as mean ± SEM (+/+ EtOH $n = 5$, +/+ TAM $n = 6$, Cre/+ EtOH $n = 7$, Cre/+TAM $n = 10$). *P* values derived from one-way ANOVA are \*\*\**p* < 0.001. **d** Flow cytometric analysis of YFP reporter gene expression 4 weeks after recombination in vivo. Data are given as means ± SEM (+/+ TAM $n = 12$, Cre/+ OIL $n = 8$, Cre/+ TAM $n = 10$). *P* values derived from one-way ANOVA are \*\*\**p* < 0.001. **e** Primary microglia from newborn *Cx3cr1CreERT2:R26-yfp, Tgfbr2fl/fl* were recombined in vitro. Control (+/+ EtOH, +/+ TAM, Cre/+ EtOH) and knockout (Cre/+ TAM) samples were treated with TGFβ1 (5 ng/ml) for 2 h. Quantification of SMAD2 phosphorylation (pSMAD2) after densitometric evaluation and GAPDH normalisation (+/+ EtOH $n = 3$, +/+ TAM $n = 3$, Cre/+ EtOH $n = 4$, Cre/+ TAM $n = 9$). Data are given as fold changes compared to +/+ EtOH-treated samples. *P* values derived from one-sample *t* tests are \**p* < 0.05 and \*\*\**p* < 0.001. **f** Nuclear translocation of SMAD2 was determined in primary microglia after treatment with TGFβ1 (5 ng/μl) for 2 h. Scale bar represents 50 μm. Data are given as means ± SEM from three independent experiments. *P* values derived from one-way ANOVA are \*\*\**p* < 0.001

## Tgfbr2 is dispensable for microglia survival and maintenance.

Body weight and survival of knockout mice as well as impaired microglia maintenance has been described in mice lacking expression of TGFβ1 in the CNS[12]. In order to evaluate the contribution of microglial TGFβ signalling to these phenotypes, body weight and survival of *Cx3cr1CreERT2:R26-yfp,Tgfbr2fl/fl* mice was analysed. Interestingly, no impairment of body weight increase up to 160 days after recombination could be observed (Fig. 2a). Moreover, mouse survival as analysed up to 315 days after recombination was not affected (Fig. 2b). Quantifications of

cortical Iba1+ microglia were performed early (60 days) and late after TAM-induced recombination (150 days). At both time points, no significant differences in microglia numbers between controls and knockout samples could be detected (Fig. 2c). However, distinct morphological changes of microglia could be observed after TAM-induced recombination (Fig. 2d, e). In order to address whether neurons numbers were affected by loss of microglial TGFβ signalling, cortical NeuN+ cells were quantified. As depicted in Fig. 2f, g, normal distribution of cortical neurons in layers 3−5 could be observed 60 days after recombination.

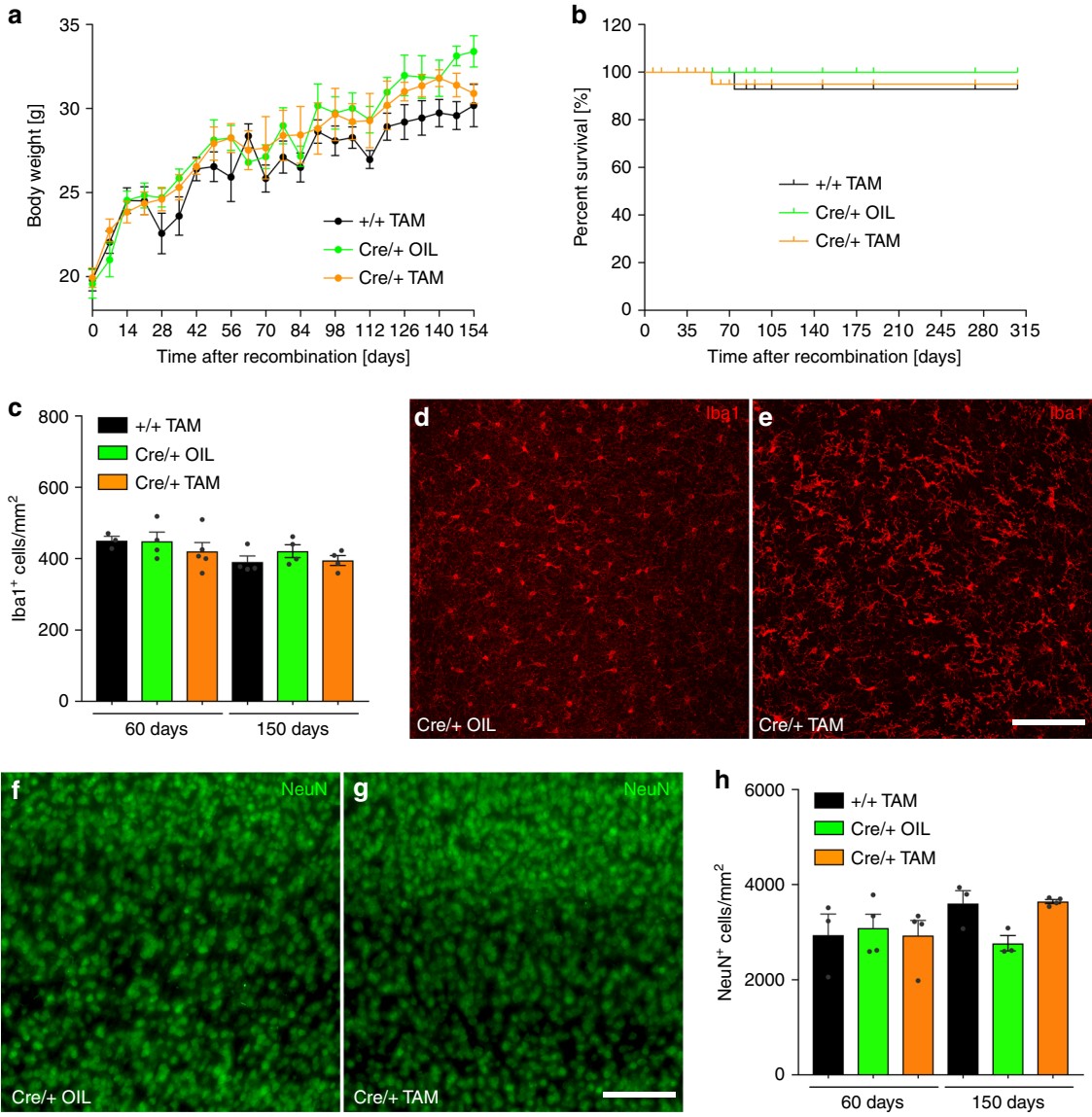

**Fig. 2** Normal mouse survival and no impairment of microglia maintenance after microglia-specific knockout of Tgfbr2. **a** Body weight increase after TAM-induced recombination at 6−8 weeks of age (timepoint 0). Weights were measured weekly in control (+/+ TAM, Cre/+ OIL) and knockout (Cre/+ TAM) animals. Data points indicate mean ± SEM ($n$ = 3–18 mice). **b** Kaplan−Meier survival curves of control and knockout mice after TAM-induced recombination in vivo. **c** Microglia numbers given as Iba1$^+$ cells/mm$^2$ were obtained from three cortical fields per animal. Iba1$^+$ microglia were analysed 60 days and 150 days after TAM-induced recombination. **d** Iba1 immunostainings from the cortices of control mice (Cre/+ OIL) and **e** knockout mice (Cre/+ TAM) after 60 days. Cortical neuron (NeuN$^+$) distributions in **f** control mice and **g** knockout mice 60 days after TAM-induced recombination. Scale bars represent 100 μm. **h** Quantification of cortical neuron numbers in control (+/+ TAM, Cre/+ OIL) and knockout (Cre/+ TAM) animals revealed no significant changes. Data are given as means ± SEM from at least three animals per genotype and timepoint

Moreover, no significant changes in neuron numbers were detected 60 days and 150 days after TAM-induced recombination (Fig. 2h). Collectively, these data indicate that TGFβ signalling is dispensable for postnatal survival and maintenance of adult microglia and does not affect the survival of cortical neurons.

***Tgfbr2* regulates microglia reactivity.** Since TGFβ1 has been demonstrated to regulate microglial homeostasis[15], we further addressed whether microglia in *Cx3cr1*$^{CreERT2}$:*R26-yfp,Tgfbr2*$^{fl/fl}$ mice displayed an activation phenotype. First, a detailed and quantitative morphometric analysis of Iba1$^+$ microglia lacking *Tgfbr2* has been performed. Control microglia (+/+ TAM and Cre/+ OIL) showed typical ramified morphologies, such as round to spindle-shaped somata and distinct arborisation patterns with

finely delineated processes (Fig. 3a, b). In contrast, YFP$^+$ *Tgfbr2*-knockout microglia (Cre/+ TAM) presented a heterogeneous morphology, which could be divided into three distinct sub-entities further referred to as "ramified", "bushy" and "hyper-trophied" microglia (Fig. 3c, d). First, the prevalence of these different microglia morphologies has been determined in all genotypes analysed. Whereas the number of ramified microglia was significantly reduced in Cre/+ TAM microglia, numbers of hypertrophied and bushy microglia were increased. Noteworthy, hypertrophied as well as bushy microglia could rarely be detected in cortices from control mice (Fig. 3e). Three-dimensional reconstructions (Fig. 3f) revealed that the filament length was reduced in hypertrophied and to a lesser extent in bushy microglia (Fig. 3g). The area covered by microglia processes was significantly reduced in both bushy and hypertrophied microglia

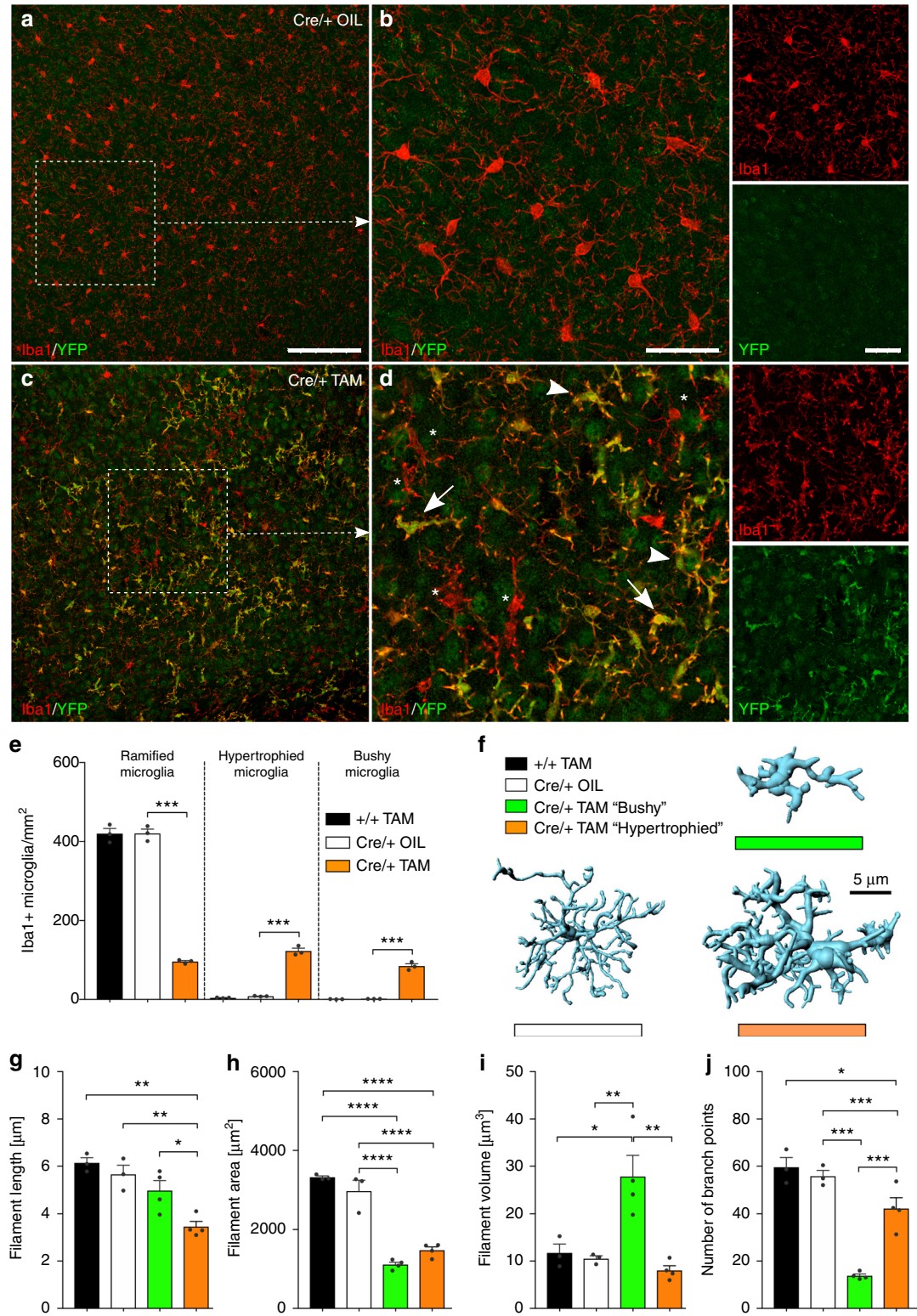

compared to controls (Fig. 3h). Filament volume was strongly increased in bushy microglia (Fig. 3i) and the numbers of branching points were dramatically reduced in bushy microglia and to a minor extent in hypertrophied microglia (Fig. 3j). Numbers of terminal points and numbers of segments (Supplementary Fig. 3) further demonstrate that *Tgfbr2*-deficient microglia display distinct morphological changes in vivo. Since morphology changes can be taken as a proxy for different

microglia activation states[23], we further analysed the gene expression patterns of *Tgfbr2*-knockout microglia by using RNAseq. Adult isolated microglia from tamoxifen-injected *Cx3cr1*$^{CreERT2}$:*R26-yfp,Tgfbr2*$^{fl/fl}$ and *Cx3cr1*$^{+/+}$:*R26-yfp,Tgfbr2*$^{fl/fl}$ mice were sorted as YFP$^+$/CD45$^{low}$ and CD11b$^+$/CD45$^{low}$, respectively (Fig. 4a). Interestingly, comparative RNAseq analysis revealed that the expression of microglia-specific genes such as *Csf1r*, *Cx3cr1*, *Fcrls*, *Olfml3*, *P2ry12*, *Sall1*, *SiglecH* or, *Tmem119*

**Fig. 3** Distinct morphological changes of microglial in *Cx3cr1^CreERT2^:R26-yfp,Tgfbr2^fl/fl^* mice. Normal microglia distribution and ramification (**a**) as well as normal microglia morphology and no YFP⁺ cells in control mice 4 weeks after TAM injections (**b**). Distinct heterogenous distribution (**c**) and morphological changes of YFP⁺/Iba1⁺ microglia 4 weeks after TAM-induced deletion of *Tgfbr2* (**d**). Based on their morphology, microglia in cre/+ TAM knockout mice were divided into "normally ramified", "bushy", and "hypertrophied". White arrowheads mark hypertrophied microglia, white arrows mark bushy microglia, and white asterisks mark YFP⁻/Iba1⁺ nonrecombined microglia. Scale bars represent 100 µm (**a**, **c**) and 25 µm (**b**, **d**). **e** Prevalence of normally ramified, bushy, and hypertrophied microglia in different experimental groups. Data are given as means ± SEM from at least three animals per genotype 4 weeks after recombination. *P* values derived from one-way ANOVA are ***$p < 0.001$. **f** Imaris (Bitplane)-based three-dimensional reconstructions of representative Iba1⁺ microglia from *Cx3cr1^CreERT2^:R26-yfp,Tgfbr2^fl/fl^* mice. Imaris-based automated quantification of microglial filament length (**g**), filament area (**h**), filament volume (**i**), and numbers of branch points (**j**). Data are presented as means ± SEM of 20 cells in 3−4 animals per group. *P* values derived from one-way ANOVA are *$p < 0.05$, **$p < 0.01$, ***$p < 0.001$ and ****$p < 0.0001$

was not impaired after TAM-induced deletion of *Tgfbr2* suggesting that TGFβ signalling is dispensable for maintenance of the molecular microglia signature in adult mice (Fig. 4b). However, analysis of differentially regulated genes in *Tgfbr2*-deficient microglia revealed 51 upregulated and 4 downregulated genes (with <2-fold change [1 log2 ratio] and $p < 0.05$; Fig. 4c). Functional categorisation using the DAVID bioinformatics database[24] for GO term enrichment analysis of biological process (Fig. 4d) molecular function (Fig. 4e) and KEGG pathway enrichment (Fig. 4f) indicate that *Tgfbr2*-deficient microglia are immunologically active. Cellular compartment enrichment analysis revealed increased expression of membrane components including *Ms4a4a*, *Ms4a14*, *Ms4a7*, *Cd74*, *Cd52*, *Mrc1*, and *Axl* as well as MHCII-related genes *H2-Aa*, *H2-Ab1*, *H2-Q5*, and *H2-Q7*. Moreover, increased expression of secreted molecules *Cp*, *Ccl7*, *Pf4*, *Tnf*, and *Tnfsf8* was detected in mutant microglia (Fig. 4g). Taken together, these data indicate that loss of TGFβ signalling in adult microglia results in a primed and activated microglia phenotype but does not impair the maintenance of the microglia-specific gene expression signature in vivo.

**Increased activation marker expression in mutant microglia.**
To address the expression of microglia activation markers in mutant mice, adult acutely isolated microglia were used for flow cytometry analysis at least 3 weeks after tamoxifen-induced recombination. Cells were gated for F4/80 and stained for activation markers CD86, CD206, CD36, and MHCII, which were chosen based on a screening for TGFβ1-regulated genes in primary microglia, where CD86, CD206, CD36 were downregulated after TGFβ treatment and upregulated after abrogation of TGFβ signalling by application of a Tgfbr1 inhibitor (Supplementary Fig. 4). In *Cx3cr1^CreERT2^:R26-yfp,Tgfbr2^fl/fl^* knockout mice, the percentages of CD86^high^ and CD206⁺ cells significantly increased in adult isolated microglia. No significant changes were observed for surface expression levels of CD36 and MHCII in *Tgfbr2*-deficient microglia (Fig. 5a, b). Analogue to in vivo experiments, primary microglia were recombined in vitro and used for flow cytometry analysis. Deletion of *Tgfbr2* in *Cx3cr1^CreERT2^:R26-yfp, Tgfbr2^fl/fl^* microglia resulted in dramatic increases of all activation markers analysed. The percentages of CD86^high^ (83.1 %), CD206⁺ (58.2 %), CD36^high^ (91.9 %), and MHCII^high^ (67.8 %) microglia in knockout samples were strongly increased with high significancies (Fig. 5c, d). We further analysed whether microglia express different activation markers at the same time. As shown in Fig. 5e, *Tgfbr2*-deficient CD86⁺ microglia were also positive for CD206 (64.5 %) and CD36 (97.1 %) and the majority of CD206⁺ microglia were double-positive for CD36 (80.0 %) indicating that microglia with abrogated TGFβ signalling increase expression of CD86, CD206 as well as CD36 (Fig. 5e). Interestingly, the observed increase in microglia surface activation markers was more pronounced in vitro suggesting that primary microglia in vitro are preactivated and/or differ in their sensitivity towards TGFβ.

**Cytokine release and TAK1 activation in *Tgfbr2*-KO microglia.**
In vitro recombined microglia were used to analyse signalling pathways involved in inflammatory responses and supernatants were collected to assess the release of chemokines and cytokines (Fig. 6a). *Tgfbr2*-deficient microglia obtained from *Cx3cr1^CreERT2^: R26-yfp,Tgfbr2^fl/fl^* mice showed a significant increase of TAK1 phosphorylation and a slight but not significant increase in p38 MAPK phosphorylation (Fig. 6b). Moreover, the secretion of CCL2 and CXCL10 was significantly increased whereas the release of CCL3 was decreased in *Tgfbr2*-deficient microglia. These observations are in accordance with CXCL10 and CCL2 expression after treatment of primary wild-type microglia with recombinant TGFβ1, which resulted in downregulation of *Cxcl10* and *Ccl2* and subsequent decrease in CXCL10 (Fig. 6c, Supplementary Fig. 5). It has been recently demonstrated that TGFβ1 is sufficient to promote microglial quiescence and inhibit IFNγ-induced microglia activation[15,25]. In order to address whether these anti-inflammatory effects of TGFβ1 are compromised, transgenic microglia from *Cx3cr1^CreERT2^:R26-yfp,Tgfbr2^fl/fl^* mice have been used to prepare primary microglia cultures for in vitro recombination. Expression of *iNos* and *Tnfα* was analysed after microglia activation with IFNγ. *Tgfbr2*-deficient microglia displayed a similar activation response upon IFNγ treatment as control microglia. However, TGFβ1 treatment did not result in inhibition of IFNγ-induced activation of *Tgfbr2*-deficient microglia (Fig. 6d, e). In summary, these data demonstrate that loss of TGFβ signalling in microglia results in increased immune cell signalling as evidenced by TAK1 phosphorylation and the release of proinflammatory cytokines CCL2 and CXCL10. Moreover, silencing TGFβ signalling renders microglia insensitive for TGFβ1-mediated inhibition of IFNγ-induced microglia activation.

**Discussion**
It has become clear that microglia activation contributes to the neuronal loss in neurodegenerative pathologies including Alzheimer's disease (AD) and Parkinson's disease (PD)[5,26]. However, during development and maintenance of the CNS microglia are constantly involved in supporting physiological functions such as synaptic pruning[27,28], synapse formation during learning[29], support of axonal outgrowth[30] as well as maintenance and survival-promoting effects for neurons[31]. In order to fulfil these physiological functions, microglia need to establish a distinct gene expression pattern which clearly distinguishes them from macrophages[10]. This microglia maturation has recently been demonstrated to take place during the first postnatal weeks in mice and is characterised by the expression of microglia-specific genes such as *P2ry12*, *Fcrls*, *Tmem119*, *Cx3cr1*, *Csf1r*, *Sall1*, *Siglech*, and *Olfml3*[10,11]. It is only partially understood which endogenous factors are involved in guiding postnatal microglia maturation. However, TGFβ1 has been identified as one important player for microglia maturation. In order to overcome the lethal phenotype of TGFβ1 knockout mice, which develop a

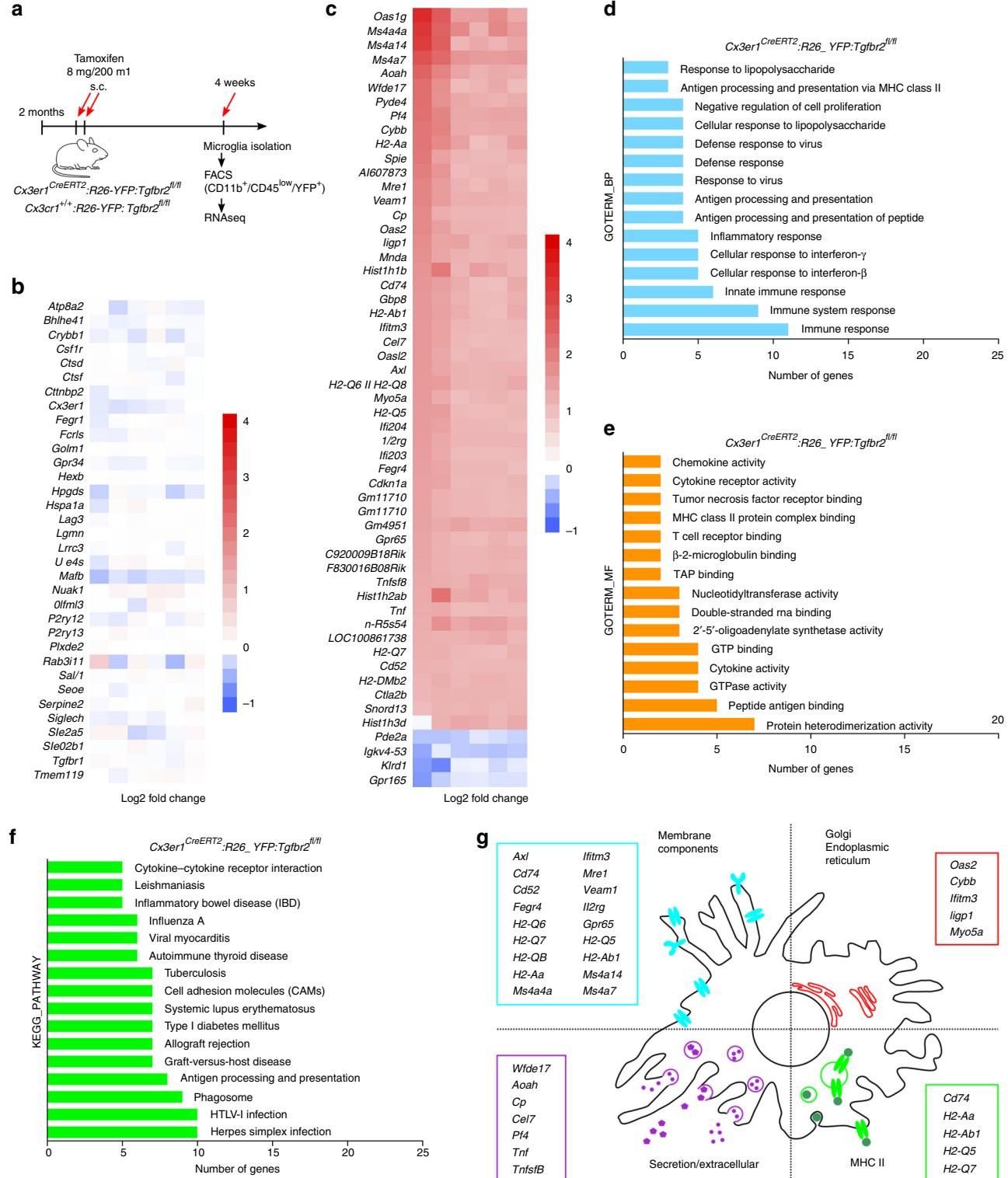

**Fig. 4** Gene expression pattern of Tgfbr2-deficient microglia indicates microglia activation and increased immune responses. **a** A timeline depicting TAM-treatment and microglia isolation/sorting for RNA isolation. **b** Heatmap analysis of RNAseq expression data of microglia isolated after TAM-induced recombination of *Cx3cr1^CreERT2^:R26-yfp,Tgfbr2^fl/fl^* mice (*n* = 6). TAM-induced *Cx3cr1^+/+^:R26-yfp,Tgfbr2^fl/fl^* mice (*n* = 6) were used as controls displaying no significant changes in expression of microglia-specific genes in *Tgfbr2*-deficient microglia. **c** Heatmap analysis of RNAseq expression data showing differentially expressed genes in *Tgfbr2*-deficient microglia. **d** GO term enrichment analysis of biological process (GOTERM_BP), **e** molecular function (GOTERM_MF) and **f** KEGG pathway enrichment analysis was performed using DAVID Bioinformatics Resources 6.8. **g** Scheme for microglia cellular compartment enrichment analysis is depicted with individually regulated compartment-associated genes

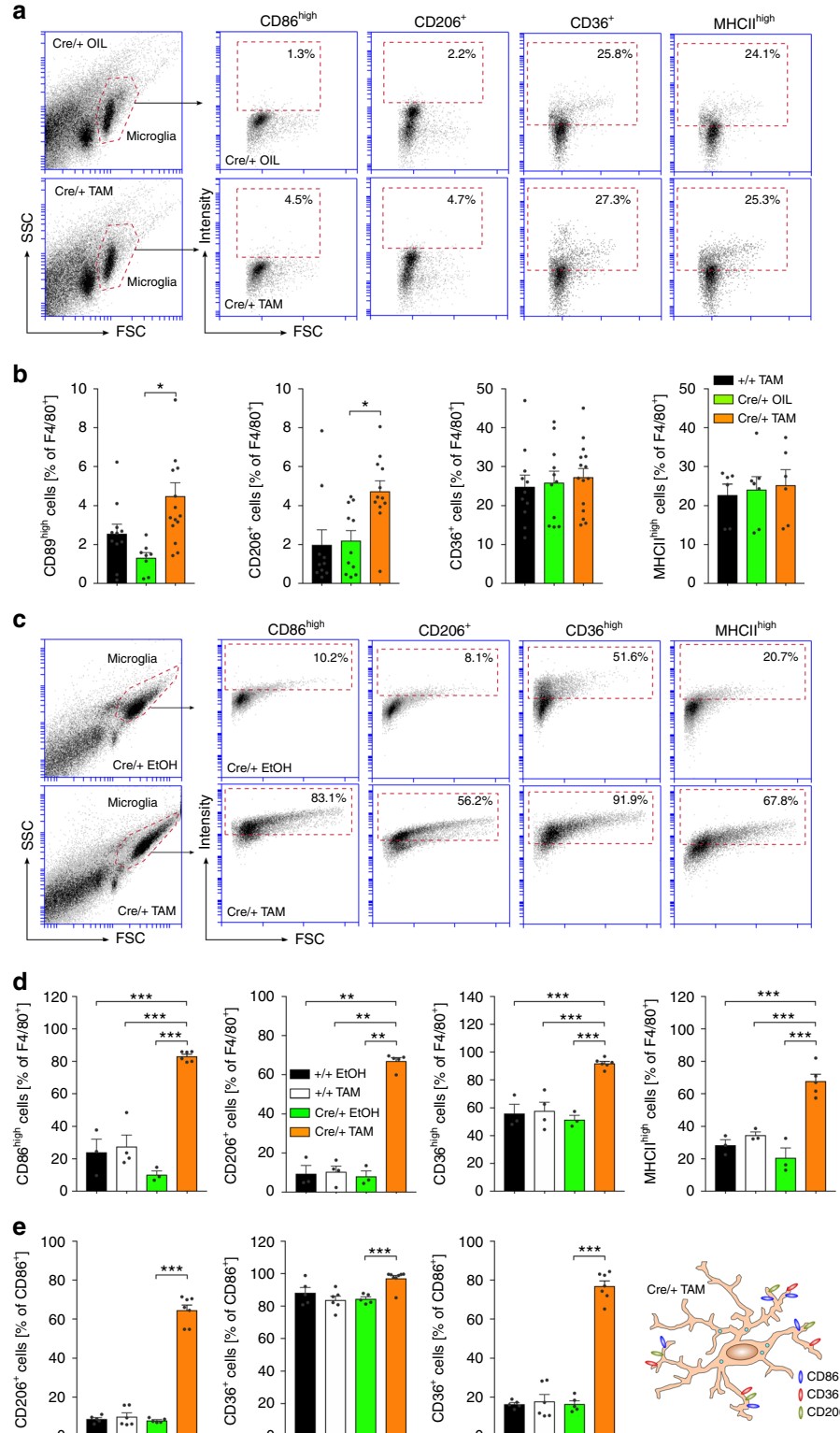

postnatal wasting syndrome due to uncontrolled systemic inflammatory responses[19], Butovsky et al.[12] reconstituted peripheral TGFβ1 expression under the control of the *Il2* promoter resulting in the survival of mice into adulthood. This CNS-specific loss of TGFβ1 lead to a microglia phenotype characterised by loss of microglia-specific gene expression patterns and impaired microglia survival starting from postnatal day 20[12]. Among the genes being affected in microglia from *TGFβ1* mutant

mice, *Sall1* has been recently reported to maintain microglia identity and their respective physiological properties[32]. Moreover, the authors used *Sall1CreERT2* mice to specifically target TGFβ signalling in adult microglia and their results suggested that TGFβ signalling in adult microglia is rather essential for microglia quiescence than for microglia maintenance and survival[32]. Using *Cx3cr1CreERT2* mice, we were able to confirm these recent findings and further demonstrate that mice undergo normal aging with

**Fig. 5** Analysis of microglial activation marker after TAM-induced deletion of *Tgfbr2* in vivo and in vitro. **a** Flow cytometry analysis of microglia activation markers CD86, CD206, CD36, and MHCII after TAM-induced recombination in vivo. Representative flow cytometry graphs are shown for microglia from Cre/+ OIL control (top) and Cre/+ TAM knockout (bottom) *Cx3cr1^CreERT2:R26-yfp,Tgfbr2^fl/fl* mice. **b** Quantifications of CD86^high, CD206^+, CD36^+ and MHCII^high microglia (F4/80^+) are depicted. Data are presented as means ± SEM of at least six independent experiments. *p* values derived from one-way ANOVA are *$p < 0.05$. **c** Flow cytometry analysis of microglia activation markers CD86, CD206, CD36 and MHCII 5 days after TAM-induced recombination in vitro. Representative flow cytometry graphs are shown for *Cx3cr1^CreERT2:R26-yfp,Tgfbr2^fl/fl* microglia after treatment with EtOH (Cre/+ EtOH, top) and TAM (Cre/+ TAM, bottom). **d** Quantifications of CD86^high, CD206^+, CD36^high and MHCII^high microglia (F4/80^+) are presented. **e** Quantifications of CD86^+/CD206^+, CD86^+/CD36^+ and CD206^+/CD36^+ double-positive microglia demonstrating the presence of all markers on the majority of F4/80^+ *Tgfbr2*-deficient microglia (Cre/+ TAM) in vitro. Data are presented as means ± SEM of at least 63 independent experiments. *p* values derived from one-way ANOVA are *$p < 0.05$, **$p < 0.01$ and ***$p < 0.001$.

regular weight gain showing unaffected microglia numbers. Using RNA-seq we were able to show that expression of microglia-specific genes was not impaired after silencing TGFβ signalling in vivo. However, in vitro studies using P0 primary microglia have clearly demonstrated that TGFβ1 regulates the expression of these microglia-specific genes (Supplementary Fig. 5D). Taken into consideration that microglia maturation—as reflected by induction and expression of microglia-specific genes[11]—peaks around P14, it is highly likely that TGFβ signalling between P0 and P14 plays a major role for this maturation process. Interestingly, we have recently demonstrated that TGFβ signalling peaks in microglia at P7, which precedes to upregulation of microglia-specific markers[33]. In the present study, male mice have been exclusively used to silence TGFβ signalling. Noteworthy, a sex dimorphism of postnatal microglia development has been described in mice[34] and thus, female mice need to be included in future studies to address whether *Tgfbr2*-deficiency results in the same phenotype in female mice. Our data further demonstrate that abrogation of TGFβ signalling in mature microglia did not result in a loss of microglia identity indicating that TGFβ is dispensable for maintenance of mature microglia. However, the molecular mechanisms and the discrimination between direct TGFβ1 target genes and genes being regulated by secondary effects remain elusive and need further evaluation in order to understand how TGFβ1 contributes to microglia maturation.

Despite the fact that microglia maturation was not impaired after silencing TGFβ signalling, we clearly demonstrated that microglia displayed an activated and primed phenotype in vivo and in vitro. Holtman et al.[35] have recently identified transcriptional profiles to distinguish between acutely activated, primed and disease-associated microglia. Comparison of these gene expression profiles with the signature observed in *Tgfbr2*-deficient microglia revealed that lack of TGFβ signalling resulted in upregulation of markers found in primed, aged, and immune-activated microglia. Among the markers suggesting microglia activation, *Cd74* and *Cd52* were upregulated in *Tgfbr2*-deficient microglia in vivo. Interestingly, treatment of microglia with recombinant TGFβ1 in vitro resulted in strong downregulation of Cd74 and Cd52 (Supplementary Fig. 4B). Both CD74 [36,37] and CD52 [38] have been reported to be upregulated in activated microglia in neuropathologic conditions. CD74, also known as the MHC class II invariant chain, acts as an MHCII chaperone and indicates increased antigen presentation. Moreover, CD74 has been identified as the receptor for MIF (macrophage migration inhibitory factor) triggering activation of ERK, MAPK, and NFκB signalling and, thus, increasing the release of proinflammatory cytokines thereby fostering immunological reactions[39]. Interestingly, the small glycoprotein CD52 has recently been shown to limit NFκB signalling in macrophages resulting in inhibition of proinflammatory cytokine production[40]. Moreover, *Cybb*, also referred to as Nox2 or gp91phox and *Tnf* were significantly upregulated after the abrogation of TGFβ signalling in

microglia. Both candidates have been extensively demonstrated to be involved in microglia-mediated neuroinflammation and neurodegeneration[41,42]. However, although several upregulated genes suggest microglia activation, several genes which have been shown to exert anti-inflammatory effects also display increased expression. *Wfdc17* as a counter-regulator of activated microglia[43], *Ms4a4a* recently described as a novel M2-like marker in macrophages[44], *Aoah* as a scavenger clearing and catabolizing free lipopolysaccharide[45], or *Mrc1* as one of the most prominent M2-like markers[46] indicate that *Tgfbr2*-deficient microglia also display anti-inflammatory properties. It is likely that these genes might be upregulated in order to compensate and counteract microglia activation and it is further unclear which of the affected genes are direct TGFβ target genes in microglia. Although some of these genes are downregulated after treatment of microglia with recombinant TGFβ1 (Supplementary Figures 4B and 5B), a thorough study identifying TGFβ target genes will gain our understanding of the TGFβ1-mediated regulation of microglia activation states. It remains further to be evaluated whether the observed microglia activation in *Cx3cr1^CreERT2:R26-yfp,Tgfbr2^fl/fl* mice impairs neuronal survival and/or functions. Numbers of cortical NeuN^+ neurons in *Cx3cr1^CreERT2:R26-yfp,Tgfbr2^fl/fl* mice were not altered compared to control mice (Fig. 2h), but effects on distinct neuron subpopulations and possible impairments of their functional properties especially during aging of mice remain to be addressed.

The involvement of TGFβ1 in the regulation of microglia activation and, thus, mediating resolution of neuroinflammation and promoting neuroprotection has been demonstrated in models for neurodegenerative diseases including AD[47,48] and PD[49]. Moreover, recent reports indicate that TGFβ1 is secreted by distinct cell populations such as astrocytes and mesenchymal stem cells to modify microglia functions[50,51]. However, the crosstalk between different CNS cells via TGFβ secretion and subsequent paracrine and/or autocrine effects is only partially understood. Whereas neurons are the major source of TGFβ1 under physiological conditions, microglia and to a lesser extent astrocytes increase expression of TGFβ1 in a middle cerebral artery occlusion model[52]. Interestingly, the levels of microglial *Tgfbr1* and *Tgfbr2* expression seem to be very low under basal conditions and rapidly increase in the same ischaemia model[53]. These observations indicate that microglia are more sensitive to TGFβ1 during their reactive states, thus, resulting in temporal and spatial limitations of TGFβ1-mediated effects on microglia. Microglia-specific conditional gene targeting has been a challenge in the past and attempts to silence TGFβ signalling using *LysM^Cre,Tgfbr2^fl/fl* mice resulted in targeted deletion in monocyte-derived cells but not in CNS resident microglia[54]. Using *Cx3cr1^CreERT2:R26-yfp,Tgfbr2^fl/fl* mice, we have clearly demonstrated an inducible microglia-specific deletion of Tgfbr2 and the fact that mutant mice do not show impaired survival and microglia maintenance offers the opportunity to study microglial TGFβ signalling in a variety of neurodegenerative disease models.

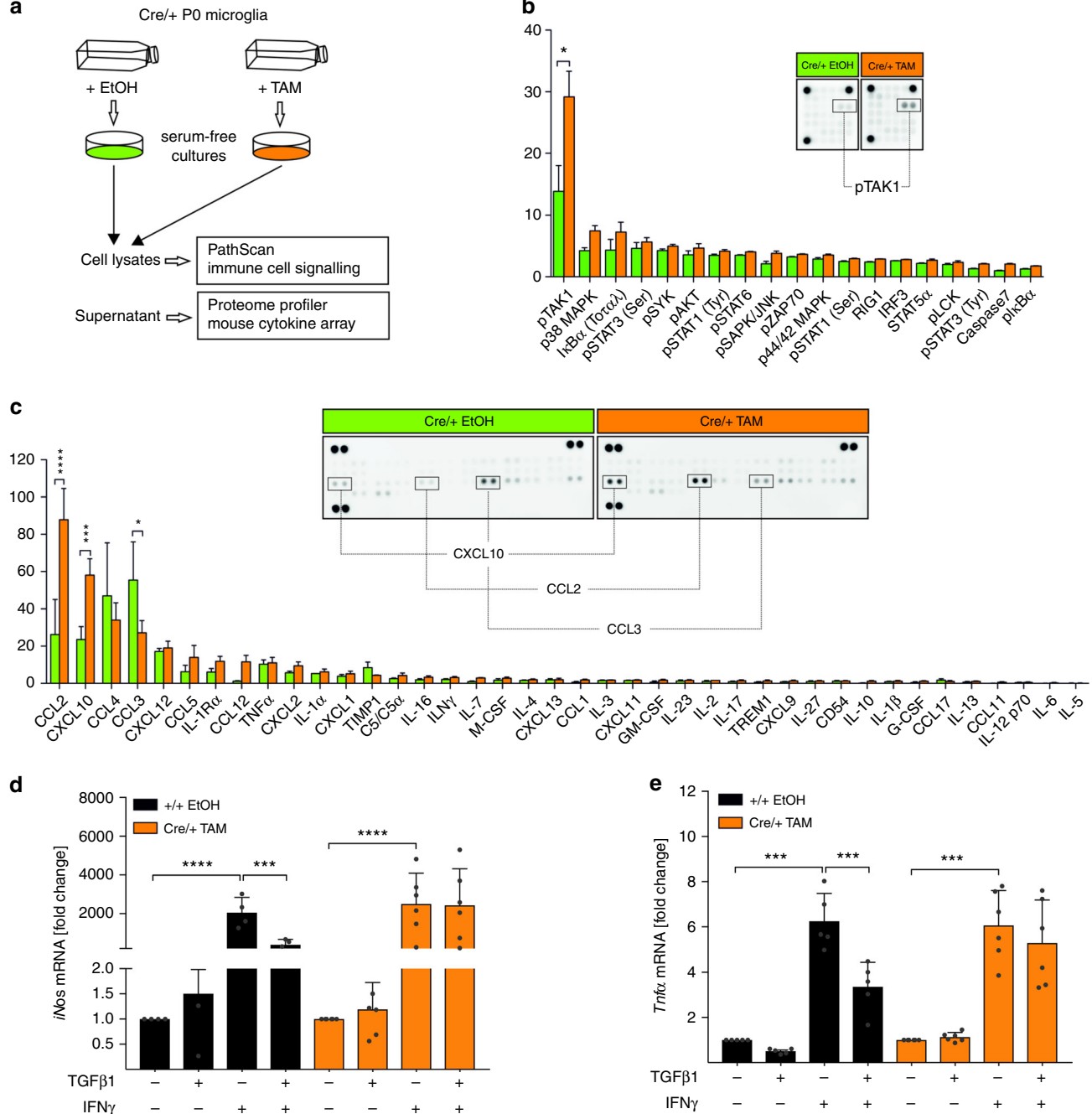

**Fig. 6** Microglial-specific knockout of *Tgfbr2* leads to increased TAK1 phosphorylation and increased CXCL10 and CCL2 secretion. **a** Primary microglia from P0 *Cx3cr1CreERT2:R26-yfp,Tgfbr2fl/fl* mice were recombined in vitro and serum-free microglia shake-off cultures were prepared after 5 days. Microglia cell lysates were used for the detection of activated signalling proteins using the CST PathScan® Immune Cell Signaling Antibody Array Kit and supernatants of control (Cre/+ EtOH) and knockout (Cre/+ TAM) microglia were used for cytokine and chemokine detection using the Proteome Profiler Mouse Cytokine Array. **b** Densitometric spot analysis of array membranes revealed increased TAK1 phosphorylation in Cre/+ TAM microglia. Data are presented as means ± SEM from eight independent experiments. *p* values derived from one-sample *t* tests are *$p < 0.05$. **c** Quantification of cytokine secretion was performed by densitometric spot analysis of array membranes. Data are presented as means ± SEM from three individual experiments. *p* values derived from one-sample *t* tests are *$p < 0.05$, ***$p < 0.001$ and ****$p < 0.0001$. Primary microglia from *Cx3cr1CreERT2:R26-yfp,Tgfbr2fl/fl* mice were recombined in vitro, stimulated with TGFβ1 (5 ng/ml) for 2 h and subsequently with IFNγ (10 ng/ml) or TGFβ1 + IFNγ for 24 h. mRNA levels of *iNos* (**d**) and *Tnfa* (**e**) values from quantitative RT-PCR were normalised to Gapdh. Graphs display fold changes compared to unstimulated samples of +/+ EtOH control samples and cre/+ TAM knockout samples, respectively. Data are presented as means ± SEM (+/+ EtOH n = 4−5, Cre/+ TAM n = 5−6). *p* values derived from one-way ANOVA are ***$p < 0.001$ and ****$p < 0.0001$

Moreover, TGFβ signalling in microglia can be further analysed at different postnatal time points including aging where impaired microglial TGFβ signalling has been suggested to be linked with age-dependent microglia activation[55]. Taken together, the present study underlines the importance of TGFβ signalling for the regulation of microglia activation in vivo and in vitro and further introduces *Cx3cr1*[CreERT2]*:R26-yfp,Tgfbr2*[fl/fl] mice as a powerful tool to study TGFβ-mediated effects during physiological and pathological functions of microglia in the CNS.

## Methods

**Mice**. The generation of *Cx3cr1*[CreERT2] mice has been described recently[22]. *Cx3cr1*[CreERT2] mice were genotyped by PCR using the forward primer 5′-CCT CTA AGA CTC ACG TGG ACC TG-3′, the reverse primer 5′-GAC TTC CGA GTT GCG GAG CAC-3′ and specification primer 5′-GCC GCC CAC GAC CGG CAA AC-3′ which amplify a 304 bp fragment from the transgenic *Cx3cr1* locus. Mice carrying *loxP*-site-flanked (floxed) alleles of the *Tgfbr2* gene[56] were crossed to the *Cx3cr1*[CreERT2] line. *R26-yfp* mice (B6.129×1-Gt(ROSA)26Sortm1(EYFP)Cos/J)[57] were additionally crossed into the *Cx3cr1*[CreERT2]*:Tgfbr2*[fl/fl] mouse line. For induction of Cre recombinase activity, 6−8-week-old *Cx3cr1*[CreERT2] mice were treated with 8 mg tamoxifen (TAM, T5648, Sigma-Aldrich, Germany) solved in 200 μl corn oil (C8267, Sigma) injected subcutaneously and intraperitoneally at two time points 48 h apart. For all experiments, littermates carrying the respective *loxP*-flanked alleles but lacking expression of Cre recombinase (+/+ TAM) or not receiving tamoxifen (cre/+ OIL) were used as controls. Male mice have been used throughout the study. All animal experiments were approved by the Federal Ministry for Nature, Environment and Consumers' Protection of the state of Baden-Württemberg and were performed in accordance to the respective national, federal and institutional regulations.

**Microglia isolation**. Mice were deeply anaesthetised by an intraperitoneal injection of ketamine (75 mg/kg) and Rompun (5.8 mg/kg) and perfused with ice-cold phosphate-buffered saline (PBS). The brains were removed from the skulls and meninges were removed by rolling over absorbent paper. Brains were collected in cold dissection buffer, homogenised with a glass homogeniser and filtered through 75 μm cell strainers. Cells were washed with 30 ml HBSS and centrifuged 12 min, 300 × *g*, 4 °C. For density gradient centrifugation, the pellet was re-suspended in 5 ml 37% Percoll (P1644, Sigma-Aldrich) in PBS, underlaid with 4 ml 70% Percoll and overlaid with 4 ml 30% Percoll in a 15 ml Falcon Tube. The gradient was centrifuged for 40 min, 600 × *g*, 15 °C, without breaks. Afterward, the cell layer was transferred to PBS+ 1% FCS and centrifuged for 5 min, 200 × *g*, 4 °C.

**Primary cultures**. Cells were prepared from newborn mice as described previously[15]. For induction of Cre recombinase activity, OH-TAM (H7904, Sigma-Aldrich, Germany) was applied at a final concentration of 1 μM at least 3 days before analysis. Microglia were kept in serum-free medium for 2 h, stimulated with TGFβ1 (5 ng/ml) for 2 h and either subsequently fixed in 4% paraformaldehyde (PFA) and processed for fluorescence microscopy or harvested in PBS for gene expression and protein analysis.

**Fluorescence microscopy**. After transcardial perfusion with PBS and 4% PFA, brains were post-fixed in 4% PFA for additional 24 h, embedded in 5% agarose, and 50 μm vibratome sections were prepared. Cultured cells were fixed with 4% PFA. Sections and cultured cells were then blocked with PBS containing 5% bovine serum albumin and permeabilized with 0.1% Triton X-100. Primary antibodies were added overnight at a dilution of 1:500 for anti-Iba1 (019-19741, WACO, Japan), 1:1000 for anti-GFP (600-106-215, Rockland Immunochemicals Inc., Gilbertsville, USA), 1:200 for anti-SMAD1/2/3 (sc-7960, Santa Cruz) and 1:200 for anti-SMAD4 (sc-7966, Santa Cruz) at 4 °C. Secondary antibodies were added as follows: Alexa Fluor 488 and Alexa Fluor 568, 1:500, for 2 h at room temperature. Nuclei were counterstained using DAPI. Iba1+ microglia were counted in at least three cortical areas (500 × 500 μm) of each animal in the maximum intensity projection of 50 μm thickness. In cultured cells, SMAD1/2/3+ and SMAD4+ microglia were counted in three visual fields at ×20 magnification. SMAD+ cells were set in relation to all DAPI+ cell. The number of cells and the examined areas were determined using a Zeiss AxioImage M2 microscope (ZEISS, Göttingen, Germany).

**Three-dimensional reconstruction of microglia**. Free-floating 50-μm vibratome sections from adult brain tissue were stained overnight with anti-IBA1 (1:500) at 4 °C, followed by Alexa Fluor 568-conjugated secondary antibody at a dilution of 1:500 for 2 h at 20−25 °C. Nuclei were counterstained with DAPI. Imaging was performed on a Leica TCS SP8 confocal laser scanning microscope with a ×20 oil immersion objective and the LAS AF image analysis software. Z-stacks with 1.1-μm steps in the z direction, 1024 × 1024 pixel resolution, were analysed using Imaris

software (Bitplane). Microglia from cortices (layers 2–5) corresponding to bregma levels −2 to −4 were used for the analysis.

**Flow cytometry**. Cells were stained with primary antibodies directed against CD11b (1:20, 53-0112-82, eBioscience), CD206 (5 μl, FAB2535C, R&D Systems), CD36 (5 μl, MCA2748A647, AbD Serotech), CD45 (1:20, 17-0451-82, eBioscience), CD86 (5 μl, MCA2463PE, AbD Serotech), F4/80 (5 μl, MCA497A488, AbD Serotech), MHCII (10 μl, MCA2401P647, AbD Serotech) at 4 °C for 15 min. An Fc receptor blocking antibody (TrueStain fcX, 101319, Biolegend) was used to avoid unspecific antibody binding. Cells were washed and analysed using a BD Accuri C6 flow cytometer or sorted by BD Cell Sorter FACS Aria Fusion and BD Cell Sorter FACS Aria III.

**qRT-PCR**. RNA of microglia was isolated using PicoPure™ RNA Isolation Kit (KIT0202, Arcturus) according to the manufacturer's protocol. Samples were treated with DNaseI (79254, Qiagen) and RNA was transcribed into cDNA using High Capacity RNA-to-cDNA Kit (4387406, Life Technologies). Five microlitres cDNA was transferred into a 96-well Multiply PCR plate (MLL 9601, Biorad) with 15 μl GoTaq® qPCR Master Mix (A6002, Promega). RT-PCR reactions were performed as described recently[33]. Primers *Tgfbr2*for 5′-TAA-CAGTGATGTCATGGCCAGCG-3′, *Tgfbr2*rev 5′-AGACTTCATGCGGCTTCTCACAGA-3′, *Gapdh*for 5′-ATGACTCTACC-CACGGCAAG-3′, *Gapdh*rev 5′-GATCTCGCTCCTGGAAGATG-3′, *iNos*for 5′-CAAGAGTTTGACCAGAGGACC-3′, *iNos*rev 5′-TGGAACCACTCG-TACTTGGGA-3′, *Tnfα*for 5′-GACCCTCACACTCAGATCAT-3′, *Tnfα*rev 5′-TTGAAGAGAACCTGGGAGTA-3′.

**Western blot analysis**. Tissues or cells were extracted in RIPA buffer (#9806, New England Biolabs). Samples were separated by SDS-PAGE and immunoblotted using antibodies to pSMAD2 (1:500, 3101s, Cell Signaling) and GAPDH (1:100, 2118, Cell Signaling). Quantification was performed using ImageJ software. Uncropped scans of the blots are presented in Supplementary Fig. 2D.

**Protein arrays**. Primary microglia were counted and plated with equal numbers on 3 cm cell culture dishes for 24 h. The cells were then washed once with PBS and given serum-free medium. After 24 h, cells were lysed and used for the PathScan® Immune Cell Signalling Antibody Array Kit (13792, Cell Signalling Technologies) and the supernatant was used for Proteome Profiler Mouse Cytokine Array Kit (ARY006, R&D) according to the manufacturer's protocols.

**RNA Seq**. Total RNA was extracted from FACS sorted microglia cells using Picopure RNA extraction kit (Life Technologies) according to the manufacturer's protocol. Isolated RNA was controlled for quantity and determination of an RNA integrity score (RIN) using RNA pico chips on a Bioanalyzer 2100 (Agilent). Sample preparation for microarray hybridisation was carried out as described in the NuGEN Ovation Pico WTA System V2 and NUGEN Encore Biotin Module manuals (NuGEN Technologies, Inc, San Carlos, CA, USA). In brief, between 0.4 and 2.5 ng of total RNA was reverse transcribed into double-stranded cDNA in a two-step process, introducing an SPIA tag sequence. Bead purified cDNA was amplified by an SPIA amplification reaction followed by an additional bead purification. 3.0 μg of SPIA cDNA was fragmented, terminally biotin-labelled and hybridised to an Affymetrix Mouse Gene 2.0 ST Array Plate. For hybridisation, washing, staining and scanning an Affymetrix GeneTitan system, controlled by the Affymetrix GeneChip Command Console software v4.2, was used. Sample processing was performed at an Affymetrix Service Provider and Core Facility, "KFB—Center of Excellence for Fluorescent Bioanalytics" (Regensburg, Germany; www.kfb-regensburg.de). Differential Gene Expression Analyses were performed using DAVID[24].

**Statistics**. All statistical analysis was performed using GraphPad Prism 6. Values are expressed as means ± standard error of the mean (SEM). Significance was assessed using a 95% confidence level. G*Power 3.1 (University Düsseldorf, Germany) was used to determine sample sizes. Unpaired, two-tailed, parametric *t* tests were used for comparison of two sets of absolute values. "One-sample *t* tests" are used for values expressed as fold changes, where the analysed value was compared to the hypothetical value 1. One-way ANOVA with correction for multiple comparisons was used for three or more sets of absolute values. Tukey multiple comparison test was used to compare all groups with each other. Two-way ANOVA was used in protein arrays to consider both changed and overall protein levels.

## Data availability

RNAseq as well as microarray data have been deposited to NCBI GEO and are available as accession numbers GSE115652 (TGFβ1-treated primary microglia, microarray data) and GSE115757 (Tgfbr2-deficient microglia, sorted, RNAseq). The data that support the findings of this study are available from the corresponding author upon reasonable request.

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

## Acknowledgements

This work was funded by grants from the Deutsche Forschungsgemeinschaft (DFG, SP 51555/2-1). The authors thank Ludmila Butenko for her excellent technical assistance.

## Author contributions

T.Z. and B.S. designed the study. T.Z., A.S., C.K., P.S.P., T.M., and B.S. performed experiments. D.P. processed RNA samples for cDNA microarrays. T.B. processed RNA samples for RNAseq. T.Z., M.P., and B.S. wrote the manuscript.

## Additional information

**Competing interests:** The authors declare no competing interests.

