## [Peer Review File · Nature Communications]

Reviewers' comments:

Reviewer #2 (Remarks to the Author):

Zoller et al. have created a mouse which allows them to silence TGFb1 signaling in CX3CR1 microglia in adult mice avoiding the lethality issues usually associated with inhibition of TGFb1 signaling. The model presented here is clearly studying TGFb signaling in microglia; whereas a different model, which knocked out TGFb1 production in the CNS was more ambiguous as to what was exactly being studied. However, that publication is highly cited and has dictated our understanding of TGFb1 signaling in microglia. This manuscript describes an important tool to be used in the study of TGFb signaling in microglia, and demonstrates that elimination of TGFb1 signaling in microglia is not required for mouse survival or microglial maintenance. However, there are a number of concerns in the presentation of the data, which make it hard to evaluate the significance of the findings:

- The use of the term quiescence to describe the microglia phenotype in the presence of TGFb1 signaling is perhaps misleading. The term is not being used in the context of cell cycle, as there are no cell cycle measurements. Perhaps the use of the word homeostatic would better serve the narrative of the manuscript, as microglia are not resting, but actively surveilling their environment.
- It would be extremely useful to have a supplemental figure where the timeline for each experiment is clearly outlined. At what age the TAM was given, duration, etc. same for the in vitro experiments. Are we looking at acute loss of TGFb signaling or chronic, and is there a difference in microglia phenotype?
- In Figure 1C, are the labels in the graph switched? Is the black bar +/- TAM? That would be more consistent with what is shown in the graphs.
- In figure 1D, it appears the top plot is mislabeled and should be labeled Cre/+ OIL? Both plots shown in Figure 1D seem to have lower percentages of YFP cells compared to the corresponding graph.
- In supplementary figure 1C, when was TAM given and when were the microglia removed? What about the Cre/+ oil control?
- Figure 3D: Text in the manuscript says filament length was reduced in both hypertrophied and bushy microglia, but the graph in the figure shows only hypertrophied microglia as effected.
- Figure 4 is the key figure in which in vivo TAM treated microglia are analyzed by RNA-sequencing, and differentially expressed genes are shown along with pathway analysis. One of the points made in the discussion is that, unlike the CNS TGFb1 knockout, this model does not show loss of the homeostatic microglia markers, however this is not shown. Figure 4 should include a panel containing the expression of the genes listed in the discussion (P2ry12, Fcrls, Tmem119, Cx3cr1, Csf1r, Sall1, Siglech and Olfml3) to demonstrate this

point. This is where the difference with the Butovsky model can be highlighted at the gene expression level; in their model these genes are all down regulated.

- Figure 4: If the authors also collected and did RNA-sequencing on the YFP-/CD45^{low} microglia, this would be a wonderful internal control for the differential gene expression.
- It would be useful to compare the microglia signature seen here with the recently reported DAM phenotype, or the signature reported by Holtman et al (2015 Acta Neuropathologica Communications). If TGF β 1 signaling is important for these neurodegenerative diseases (as suggested in the discussion), then it would be interesting to see how the expression profiles align. Are they highly conserved, or only share a few genes, such as Axl, or specific pathways? Does inhibiting TGF β 1 signaling recapitulate the phenotype induced by disease? Here it would also be useful to have RNA-sequencing or even targeted qPCR to examine if the phenotype changes over time. Does Age + loss of TGF β signaling = neurodegenerative microglia phenotype? Partially or completely?
- In the discussion, it is mentioned that cortical NeuN⁺ neuron numbers are not altered, this should be shown, and when the mice are aged are they still unaffected?

Reviewer #3 (Remarks to the Author):

This study was performed to determine the role of TGF-beta signaling within adult microglia in the adult mouse brain. Young adult mice were injected with tamoxifen to induce Cre-mediated deletion of TGF β 2 in Cx3cr1-Cre-ER expressing cells, and experiments performed 60 and 150 days post recombination. Interestingly, the authors determined that TGF β 2 signaling is not required for microglial survival in adult animals, but microglia are nevertheless morphologically and transcriptionally very different in the absence of TGF β 2 signals, and display different cell surface proteins and receptors. However, no functional consequences of this changed transcriptome were explored, and in fact cells in vitro responded similarly to external challenge with IFN γ . In light of this, the discussion is overly speculative with regards to how microglia that lack TGF β 2 will respond to external stimuli.

Main specific comments follow:

1. Cx3cr1 is expressed in microglia and other tissue-resident macrophages, but it is also expressed by circulating monocytes. While the time course of tamoxifen exposure and subsequent ageing of the mice should ensure complete turnover of monocyte population, the authors should verify minimal recombination in peripheral monocytes by flow cytometry.
2. The sex of the animals used in all the experiments should be mentioned.
3. Since the recombination rates appear to be approximately 80%, this provides a unique opportunity to study the role of TGF-beta signaling. The authors note two different

morphologies in the Cre/+ TAM animals, but do not specify the morphology of non-recombined cells within these animals. Are the non-recombined cells morphologically similar to +/+ TAM exposed animals, or are they also divided into "bushy" and "hypertrophied" phenotypes?

4. Whether "bushy" and "hypertrophied" phenotypes were observed (with reduced numbers perhaps) in control conditions is unclear. The prevalence of "normally-ramified", "bushy" and "hypertrophied" microglial cells should be determined in the different experimental groups. How the different morphological phenotypes were identified should be explained. Considering that microglia are heterogeneous, the sample size of 9 cells per experimental condition (from 3-4 animals) also appears insufficient. The cortical regions and layers selected for analysis should additionally be mentioned, together with the Bregma levels used.

5. The confusion of seeming increases in pro-inflammatory and anti-inflammatory markers seen in RNAseq could be explained by the two different morphological phenotypes. Do the morphologically distinct "bushy" and "hypertrophied" microglia express different levels of cell surface markers such as CD86 or CD206? It would be useful to determine if these phenotypes could be differentiated from one another and studied independently.

6. More comprehensive flow cytometry experiments should be performed to determine if the same subset of microglia elevate the markers displayed (i.e. are the CD86^{high} cells also CD206⁺?)

7. How functionally relevant is the supposed "primed," state of TGF β 2^{-/-} microglia? Cultured cells both respond with vigor to IFN γ stimulation as shown in figure 6D and E. Do mice lacking TGF β 2 also respond to challenges in the same way? A cohort of mice should be subjected to immune challenges or a disease model to how functionally relevant this signaling is. Do these mice develop behavioral abnormalities? Behavioral tests should be performed to determine if there are any differences between these mice as well.

Minor suggestions:

8. The title of the paper has redundancy. It could be simplified by "Silencing of TGF β signalling in microglia results in impaired quiescence".

9. Microglial "activation" is now considered an outdated terminology, considering that microglia undergo various adaptive changes depending on the pathophysiological context. This appellation should be replaced by more specific information regarding their functional changes.

10. "Since morphology can be taken as a proxy for microglia activation²⁰"

There is evidence that anti-inflammatory microglia can show enlarged soma and reduced process arborization, and vice versa.

11. On page 7, the link between the molecular changes and microglial activation or priming is not clear. How these changes relate to microglial functions should be explained. Activation and priming would need to be defined also.

12. On page 8, what is meant by "surface activation" markers should be explained and the molecular changes linked to microglial functions.

13. In the Discussion, "Both Cd74^{31,32} and Cd52³³ have been reported to be upregulated in activated microglia in neuropathologic conditions."

It would be important to discuss about the function of Cd74 and Cd52.

14. On page 12: "Interestingly, the levels of microglial Tgf β 1 and Tgf β 2 expression seem

to be very low under basal conditions and rapidly increase the same ischemia model⁴⁶." A word is missing in that sentence.

15. On page 12: the abbreviations AD and PD are provided, but not used subsequently.

16. In the Methods, "For induction of Cre recombinase activity, 6-8- week-old Cx3cr1CreERT2 mice were treated with 8/16 mg tamoxifen (TAM, T5648, Sigma-Aldrich, Germany) solved in 200/400 µl corn oil (C8267, Sigma) injected subcutaneously and intraperitoneally at two time points 48 h apart."

The dose of tamoxifen and route of administration used in the study is not clear from that sentence.

17. On page 14: "sculls" should read "skulls".

18. On page 15, Fluorescence microscopy section: how the brains were embedded is not clear.

19. On page 18, Statistics section: posthoc analyses should be mentioned.

Response to Referees Letter

The authors thank the reviewer's for their constructive comments and suggestion. All changes made to the original manuscript are marked in red.

Reviewer #2 (Remarks to the Author):

Zoller et al. have created a mouse which allows them to silence TGFb1 signaling in CX3CR1 microglia in adult mice avoiding the lethality issues usually associated with inhibition of TGFb1 signaling. The model presented here is clearly studying TGFb signaling in microglia; whereas a different model, which knocked out TGFb1 production in the CNS was more ambiguous as to what was exactly being studied. However, that publication is highly cited and has dictated our understanding of TGFb1 signaling in microglia. This manuscript describes an important tool to be used in the study of TGFb signaling in microglia, and demonstrates that elimination of TGFb1 signaling in microglia is not required for mouse survival or microglial maintenance. However, there are a number of concerns in the presentation of the data, which make it hard to evaluate the significance of the findings:

- The use of the term quiescence to describe the microglia phenotype in the presence of TGFb1 signaling is perhaps misleading. The term is not being used in the context of cell cycle, as there are no cell cycle measurements. Perhaps the use of the word homeostatic would better serve the narrative of the manuscript, as microglia are not resting, but actively surveilling their environment.

Response: We thank the reviewer for all valuable comments and suggestions. We agree, that the term quiescence might be misleading due to its use in the context of cell cycle regulations and have changed this term throughout the manuscript. Homeostasis is now used instead of quiescence.

- It would be extremely useful to have a supplemental figure where the timeline for each experiment is clearly outlined. At what age the TAM was given, duration, etc. same for the in vitro experiments. Are we looking at acute loss of TGFb signaling or chronic, and is there a difference in microglia phenotype?

Response:

Detailed timelines displaying experimental approaches *in vivo* and *in vitro* have been added to Supplementary Figure 1. The results in the present study are describing the phenotype after acute loss of TGFb signalling. The effects of chronic loss of TGFb signalling are currently under investigation in a project addressing normal aging of mutant mice up to 24 months.

- In Figure 1C, are the labels in the graph switched? Is the black bar +/- TAM? That would be more consistent with what is shown in the graphs.

Response: Indeed, the labels in the representative flow dot plots have been accidentally switched. The labelling of the graph bars are correct. The labels have been corrected in the revised Figure 1.

- In figure 1D, it appears the top plot is mislabeled and should be labeled Cre/+ OIL? Both plots shown in Figure 1D seem to have lower percentages of YFP cells compared to the corresponding graph.

Response: The mislabelling of the plot in Fig.1D has been corrected. Moreover, suitable

representative dot plots have been chosen matching to the means of YFP⁺ cells.

• In supplementary figure 1C, when was TAM given and when were the microglia removed? What about the Cre/+ oil control?

Response: We have added detailed time lines to supplementary figure 1 describing *in vivo* and *in vitro* experiments. TAM was given at the age of 2 months and microglia were isolated after 4 weeks. We have also tested microglia from Cre/+ OIL treated microglia and found similar expression compared to wt microglia. These data have been added to supplementary figure 1.

• Figure 3D: Text in the manuscript says filament length was reduced in both hypertrophied and bushy microglia, but the graph in the figure shows only hypertrophied microglia as effected.

Response: Correct, the filament length was significantly reduced in hypertrophied microglia and to a lesser (non significant) extent in bushy microglia. This has been corrected in the result section accordingly.

• Figure 4 is the key figure in which *in vivo* TAM treated microglia are analyzed by RNA-sequencing, and differentially expressed genes are shown along with pathway analysis. One of the points made in the discussion is that, unlike the CNS TGFB1 knockout, this model does not show loss of the homeostatic microglia markers, however this is not shown. Figure 4 should include a panel containing the expression of the genes listed in the discussion (P2ry12, Fcrls, Tmem119, Cx3cr1, Csf1r, Sall1, Siglech and Olfml3) to demonstrate this point. This is where the difference with the Butovsky model can be highlighted at the gene expression level; in their model these genes are all down regulated.

Response: The authors thank the reviewer for this valuable comment. We have included a heatmap depicting the expression of microglia-specific genes as Fig. 4B. We hope that this modification underlines the difference between our transgenic model to the previously reported mouse line presented by the Butovsky group.

• Figure 4: If the authors also collected and did RNA-sequencing on the YFP-/CD45^{low} microglia, this would be a wonderful internal control for the differential gene expression.

Response: We have not collected the YFP- microglia from Tgfr2 mutant mice. Indeed, this would have been an interesting internal control. The changes in expression of depicted genes was calculated based on TAM-treated Cx3cr1^{+/+};R26-yfp,Tgfr2^{fl/fl} microglia. We have further collected RNA from OIL-treated Cx3cr1^{CreERT2};R26-yfp,Tgfr2^{fl/fl} microglia and compared their gene expression, but did not observe changes compared to wild type microglia.

• It would be useful to compare the microglia signature seen here with the recently reported DAM phenotype, or the signature reported by Holtman et al (2015 Acta Neuropathologica Communications). If TGFb1 signaling is important for these neurodegenerative diseases (as suggested in the discussion), then it would be interesting to see how the expression profiles align. Are they highly conserved, or only share a few genes, such as Axl, or specific pathways? Does inhibiting TGFb1 signaling recapitulate the phenotype induced by disease? Here it would also be useful to have RNA-sequencing or even targeted qPCR to examine if the phenotype changes over time. Does Age + loss of TFGb signaling = neurodegenerative microglia phenotype? Partially or completely?

Response: Thank you very much for this suggestion. We have used the transcriptional profiles described by Holtman and colleagues and compared these signatures with the gene expression profile of Tgfr2-deficient microglia. We have observed that there is only a partial alignment

with aged, primed or immune activated microglia. Tgfbr2-deficient microglia show regulated genes from all transcriptional consensus profiles. At this point it remains elusive whether different microglia subpopulation are present in our mutant mice. A very interesting aspect that needs to be addressed in future studies. At this point we can not give sufficient information to address this question. However, we have started a large follow-up study to analyse the effects of lack of TGF β signalling for normal aging (up to 24 months) in combination with neurodegenerative disease models. We hope that this study will gain our understanding of the importance of TGF β signalling for adult microglia.

• In the discussion, it is mentioned that cortical NeuN⁺ neuron numbers are not altered, this should be shown, and when the mice are aged are they still unaffected?

Response: We have added the quantification of cortical NeuN⁺ neurons to Fig.2 and elaborated the results section accordingly. We have analysed neuron numbers 60 days and 150 days after recombination. To this point we can not comment on the development of neuron numbers in different functional brain during aging of mutant. This follow-up study has recently been started to address the effect of loss of microglial TGF β signalling for age-dependent neuron survival and functional outcome. The study is designed to obtain data from aged mice up to 24 months.

Reviewer #3 (Remarks to the Author):

This study was performed to determine the role of TGF-beta signaling within adult microglia in the adult mouse brain. Young adult mice were injected with tamoxifen to induce Cre-mediated deletion of TGFbr2 in Cx3cr1-Cre-ER expressing cells, and experiments performed 60 and 150 days post recombination. Interestingly, the authors determined that TGFbr2 signaling is not required for microglial survival in adult animals, but microglia are nevertheless morphologically and transcriptionally very different in the absence of TGFbr2 signals, and display different cell surface proteins and receptors. However, no functional consequences of this changed transcriptome were explored, and in fact cells in vitro responded similarly to external challenge with IFN γ . In light of this, the discussion is overly speculative with regards to how microglia that lack TGFbr2 will respond to external stimuli.

Main specific comments follow:

1. Cx3cr1 is expressed in microglia and other tissue-resident macrophages, but it is also expressed by circulating monocytes. While the time course of tamoxifen exposure and subsequent ageing of the mice should ensure complete turnover of monocyte population, the authors should verify minimal recombination in peripheral monocytes by flow cytometry.

Response: We thank the reviewer for this valuable comment. We have analysed the tamoxifen-induced recombination in blood CD115⁺/Ly6C^{low} blood monocytes 1 week and 4 weeks after tamoxifen injections. Whereas 43% YFP⁺ monocytes could be detected after 1 week, only 9.8% YFP⁺ cells were observed 4 weeks after tamoxifen applications. Recombination efficacy of microglia remained stable around 84% at both time points analysed. We have added this information in the results section of the manuscript and generated a new supplementary figure (Fig. S2).

2. The sex of the animals used in all the experiments should be mentioned.

Response: In order to exclude TAM-mediated effects in female mice, we have used only included male mice in the current study. This important information has been added to the methods section of the manuscript.

3. Since the recombination rates appear to be approximately 80%, this provides a unique opportunity to study the role of TGF-beta signaling. The authors note two different morphologies in the Cre/+ TAM animals, but do not specify the morphology of non-recombined cells within these animals. Are the non-recombined cells morphologically similar to +/+ TAM exposed animals, or are they also divided into “bushy” and “hypertrophied” phenotypes?

Response: We have observed that non-recombined cells display a normal ramification in most cases. Interestingly, some non-recombined cells in close proximity to bushy and/or hypertrophied microglia adopt also different morphologies. It is highly likely that paracrine influences of mutant microglia result in stimulation or activation of non-recombined cells. However, this phenomenon needs further experimental validation to comment on the underlying mechanisms.

4. Whether “bushy” and “hypertrophied” phenotypes were observed (with reduced numbers perhaps) in control conditions is unclear. The prevalence of “normally-ramified”, “bushy” and “hypertrophied” microglial cells should be determined in the different experimental groups. How the different morphological phenotypes were identified should be explained. Considering that microglia are heterogeneous, the sample size of 9 cells per experimental condition (from 3-4 animals) also appears insufficient. The cortical regions and layers selected for analysis should additionally be mentioned, together with the Bregma levels used.

Response: We agree to the reviewer's comment and have added the prevalence of normally ramified, bushy and hypertrophied microglia in different experimental groups. Microglia were categorized in these three groups and counted based on the presence of extensive ramification, their process thickness and "bushy" microglia were identified due to the lack of proper ramification. We know that this classification is a rough method for the description of microglia morphology and that we can not comment or conclude on the activation status of these different microglia forms. We have further increased the numbers of cells analyzed using IMARIS to 20 cells per mice. The statistics for the parameters which have been analyzed have not changed. Figure 3 has been updated and revised accordingly. The information about the cortical layers and the bregma levels have been added to the materials and methods section.

5. The confusion of seeming increases in pro-inflammatory and anti-inflammatory markers seen in RNAseq could be explained by the two different morphological phenotypes. Do the morphologically distinct “bushy” and “hypertrophied” microglia express different levels of cell surface markers such as CD86 or CD206? It would be useful to determine if these phenotypes could be differentiated from one another and studied independently.

Response: Indeed, a very interesting and important point. We were also surprised to see pro- and anti-inflammatory markers upregulated at the same time. Based on our experiments, we can conclude that microglia in vitro express both marker types in parallel. A phenomenon that has also been reported in vivo. At the moment we can not comment to which extent bushy and hypertrophied microglia represent different subpopulation in vivo. Further, we don't know whether these morphology states are stable or whether hypertrophied microglia adopt bushy morphologies and vice versa. We are currently planning to use two-photon imaging of YFP⁺ microglia in vivo to see the temporal dynamics of these different morphologies. A more sophisticated differentiation between both subtypes and the functional relevance linked to these

morphologies are some of the challenging questions of the follow-up study. We also believe that classical "activation markers" are only partially useful to describe the microglia reaction states *in vivo*. We are currently elaborating the palette of markers (also based on RNAseq data) to better understand what microglia functional changes are triggered by loss of TGFb signalling *in vivo*.

6. More comprehensive flow cytometry experiments should be performed to determine if the same subset of microglia elevate the markers displayed (i.e. are the CD86^{high} cells also CD206⁺?)

Response: We have added informations about double-positivity for different activation markers to Fig. 5E. Since only a very slight change in activation marker expression was observed *in vivo*, we have performed this analysis with the *in vitro* recombined microglia. According to these experiments a relatively homogenous microglia population which displays positivity for CD86, CD206, and CD36 could be observed. A sophisticated follow-up study addressing expression of activation markers in neurodegenerative disease models *in vivo* is currently being conducted where simultaneous expression of several activation markers is analyzed.

7. How functionally relevant is the supposed "primed," state of TGFb2^{-/-} microglia? Cultured cells both respond with vigor to IFN γ stimulation as shown in figure 6D and E. Do mice lacking TGFb2 also respond to challenges in the same way? A cohort of mice should be subjected to immune challenges or a disease model to how functionally relevant this signaling is. Do these mice develop behavioral abnormalities? Behavioral tests should be performed to determine if there are any differences between these mice as well.

Response: We appreciate this important comment. The observation that mutant microglia respond similar to IFN γ stimulation *in vitro* suggests that the extent of the reaction is not impaired by loss of TGFb signalling *in vitro*. However, *in vitro* microglia are well known to be pre-activated, a fact that likely affects this observation. Interestingly, the ability of TGFb1 to inhibit IFN γ -induced microglia activation is lost in mutant cells. We have started an extensive follow-up study to analyse these mice (and conditional Smad4-mutant mice) in neuroinflammatory models *in vivo*. We are convinced that the results from this study will elucidate the functional relevance of microglial TGFb signalling *in vivo*.

Minor suggestions:

8. The title of the paper has redundancy. It could be simplified by "Silencing of TGFb signalling in microglia results in impaired quiescence".

Response: The title of the paper has been changed in accordance to the reviewer's suggestion. Thank you for this valuable suggestion.

9. Microglial "activation" is now considered an outdated terminology, considering that microglia undergo various adaptive changes depending on the pathophysiological context. This appellation should be replaced by more specific information regarding their functional changes.

Response: We agree to the important comment that microglia activation is an outdated terminology, especially with regard to M1 or M2 activation states. We have updated the introduction accordingly highlighting this aspect. Moreover, we have tried to avoid using "microglia activation" as a general description of a reactive state. However, the upregulated genes we have observed in our model neither completely fit to primed microglia or aged microglia, nor to acutely activated (e.g. LPS challenged) microglia. Thus, we kept the term activation in some cases since we observed an activation without knowing the trigger of this event.

10. "Since morphology can be taken as a proxy for microglia activation²⁰"

There is evidence that anti-inflammatory microglia can show enlarged soma and reduced process arborization, and vice versa.

Response: We completely agree that morphological changes can not be used to draw conclusions about the activation states of microglia. However, any change in morphology is due a stimulating trigger. Thus, we have changed the sentence in order to emphasize that morphology changes are likely to indicate different activation states, which should further be validated using gene expression analyses.

11. On page 7, the link between the molecular changes and microglial activation or priming is not clear. How these changes relate to microglial functions should be explained. Activation and priming would need to be defined also.

Response: We apologize for the missing definitions. Microglia activation and priming as well as the functional consequences have been defined in an additional section of the introduction part of the manuscript. The references have been updated accordingly.

12. On page 8, what is meant by "surface activation" markers should be explained and the molecular changes linked to microglial functions.

Response: The sentence was intended to describe that activation markers located on the outer surface of microglia were used for the analysis. We have rephrased this sentence in order to reduce the confusion. The molecular/microglia functions linked to these markers are discussed in the discussion section.

13. In the Discussion, "Both Cd74^{31,32} and Cd52³³ have been reported to be upregulated in activated microglia in neuropathologic conditions."

It would be important to discuss about the function of Cd74 and Cd52.

Response: We have added informations about the functions of CD74 and CD52 to the discussion section of the manuscript.

14. On page 12: "Interestingly, the levels of microglial Tgfbr1 and Tgfbr2 expression seem to be very low under basal conditions and rapidly increase the same ischemia model⁴⁶."

A word is missing in that sentence.

Response: The missing word "in" has been added to the sentence.

15. On page 12: the abbreviations AD and PD are provided, but not used subsequently.

Response: We have revised the manuscript and introduced the abbreviations the first time Alzheimer's disease and Parkinson's disease are mentioned and further used these abbreviations.

16. In the Methods, "For induction of Cre recombinase activity, 6-8- week-old Cx3cr1CreERT2 mice were treated with 8/16 mg tamoxifen (TAM, T5648, Sigma-Aldrich, Germany) solved in 200/400 µl corn oil (C8267, Sigma) injected subcutaneously and intraperitoneally at two time points 48 h apart."

The dose of tamoxifen and route of administration used in the study is not clear from that sentence.

Response: We are sorry for this inconvenience. We have updated the material and methods section describing the subcutaneous injection of 8mg tamoxifen/200 µl corn oil at two time points 48 h apart.

17. On page 14: "sculls" should read "skulls".

Response: "sculls" has been changed to "skulls".

18. On page 15, Fluorescence microscopy section: how the brains were embedded is not clear.

Response: The missing information how PFA-fixed brains were embedded in 5% agarose to perform vibratome sections have been added to the Fluorescence microscopy section.

19. On page 18, Statistics section: posthoc analyses should be mentioned.

Response: We are sorry for this missing information. We have added this information (Tukey's multiple comparison test) to the statistics section.

REVIEWERS' COMMENTS:

Reviewer #2 (Remarks to the Author):

The authors have addressed all of my comments appropriately.

Reviewer #3 (Remarks to the Author):

Holler and colleagues did an excellent job at addressing most of the reviewers comments, and provided good explanations of follow-up studies in the cases they didn't address our comments in the manuscript. My biggest major concern previously was the small N for morphological studies, which they addressed very well.

I'm still concerned about the use of only male mice for the studies and think the authors should spend a sentence or two in the discussion to address that point, especially as they are marking TGFb peak expression at P7 as a developmental milestone. They should mention the caveat of different developmental timelines of male VS female microglia (review here: <https://www.ncbi.nlm.nih.gov/pubmed/24871624>).

Response to REVIEWERS' COMMENTS:

Reviewer #2 (Remarks to the Author):

The authors have addressed all of my comments appropriately.

Reviewer #3 (Remarks to the Author):

Holler and colleagues did an excellent job at addressing most of the reviewers comments, and provided good explanations of follow-up studies in the cases they didn't address our comments in the manuscript. My biggest major concern previously was the small N for morphological studies, which they addressed very well.

I'm still concerned about the use of only male mice for the studies and think the authors should spend a sentence or two in the discussion to address that point, especially as they are marking TGFb peak expression at P7 as a developmental milestone. They should mention the caveat of different developmental timelines of male VS female microglia (review here: <https://www.ncbi.nlm.nih.gov/pubmed/24871624>).

Response: The authors thank the reviewer for this valuable comment. We agree that sex differences during microglia development need to be discussed and addressed in the follow-up study using Tgfr2-deficient mice. An appropriate section has been added to the discussion.